# Characterization of Glossy Spike Mutants and Identification of Candidate Genes Regulating Cuticular Wax Synthesis in Barley (*Hordeum vulgare* L.)

**DOI:** 10.3390/ijms232113025

**Published:** 2022-10-27

**Authors:** Xiuxiu Bian, Lirong Yao, Erjing Si, Yaxiong Meng, Baochun Li, Xiaole Ma, Ke Yang, Yong Lai, Xunwu Shang, Chengdao Li, Juncheng Wang, Huajun Wang

**Affiliations:** 1College of Agronomy, Gansu Agricultural University, Lanzhou 730070, China; 2State Key Lab of Aridland Crop Science/Gansu Key Lab of Crop Improvement and Germplasm Enhancement, Gansu Agricultural University, Lanzhou 730070, China; 3College of Life Sciences and Technology, Gansu Agricultural University, Lanzhou 730070, China; 4College of Forestry, Henan Agricultural University, Zhengzhou 450002, China; 5Western Crop Genetic Alliance, College of Science, Health, Engineering and Education, Murdoch University, 90 South Street, Murdoch, WA 6150, Australia

**Keywords:** cuticular waxes, barley, gene, 16-hydroxyhexadecanoic acid, synthesis

## Abstract

Cuticular waxes comprise the hydrophobic layer that protects crops against nonstomatal water loss and biotic and abiotic stresses. Expanding on our current knowledge of the genes that are involved in cuticular wax biosynthesis and regulation plays an important role in dissecting the processes of cuticular wax metabolism. In this study, we identified the Cer-GN1 barley (*Hordeum vulgare* L.) mutant that is generated by ethyl methanesulfonate mutagenesis with a glossy spike phenotype that is controlled by a single recessive nuclear gene. A physiological analysis showed that the total cuticular wax loads of Cer-GN1 were one-third that of the progenitor wild-type (WT), and its water loss rate was significantly accelerated (*p* < 0.05). In addition, Cer-GN1 was defective in the glume’s cuticle according to the toluidine blue dye test, and it was deficient in the tubule-shaped crystals which were observed on the glume surfaces by scanning electron microscopy. Using metabolomics and transcriptomics, we investigated the impacts of cuticular wax composition and waxy regulatory genes on the loss of the glaucous wax in the spikes of Cer-GN1. Among the differential metabolites, we found that 16-hydroxyhexadecanoic acid, which is one of the predominant C16 and C18 fatty acid-derived cutin monomers, was significantly downregulated in Cer-GN1 when it was compared to that of WT. We identified two novel genes that are located on chromosome 4H and are downregulated in Cer-GN1 (*HvMSTRG.29184* and *HvMSTRG.29185*) that encode long-chain fatty acid omega-monooxygenase CYP704B1, which regulates the conversion of C16 palmitic acid to 16-hydroxyhexadecanoic acid. A quantitative real-time PCR revealed that the expression levels of *HvMSTRG.29184* and *HvMSTRG.29185* were downregulated at 1, 4, 8, 12, and 16 days after the heading stage in Cer-GN1 when it was compared to those of WT. These results suggested that *HvMSTRG.29184* and *HvMSTRG.29185* have CYP704B1 activity, which could regulate the conversion of C16 palmitic acid to 16-hydroxyhexadecanoic acid in barley. Their downregulation in Cer-GN1 reduced the synthesis of the cuticular wax components and ultimately caused the loss of the glaucous wax in the spikes. It is necessary to verify whether *HvMSTRG.29184* and *HvMSTRG.29185* truly encode a CYP704B1 that regulates the conversion of C16 palmitic acid to 16-hydroxyhexadecanoic acid in barley.

## 1. Introduction

Plant growth and development are affected by various abiotic and biotic stresses, such as droughts, high salinity conditions, pathogens, and pests, which result in significantly lower plant growth rate and crop productivity [1]. In higher plants, the outer epidermal surface of most of the above-ground tissues is covered by the cuticle, which forms the first barrier of defense against environmental threats [2]. The cuticle plays an important role in controlling nonstomatal water loss [3] and protecting the plants from pathogens [4], insect attacks [5], and UV radiation [6]. Although the ultrastructure and composition of the cuticle differ among the plant species, organs, developmental stages, and environmental cues, the capacity to synthesize the hydrophobic surface layers is evolutionarily conserved among the land plants [7]. In most of the plant species, the cuticular wax consists of very long-chain fatty acids (VLCFAs; C20–C34) and their derivatives, including alkanes, aldehydes, primary and secondary alcohols, ketones, and esters [REF]. There are also some lipophilic secondary metabolites in the compounds, such as pentacyclic triterpenoids, flavonoids, and tocopherols [8,9,10].

Long-chain fatty acyl groups (C16 or C18) are generated by the fatty acid synthase (FAS) complex using acetyl-coenzyme A (CoA) substrates in the plastid of epidermal cells, and they are released by an acyl-ACP thioesterase (FATB) [11]. The fatty acids are then converted to fatty acetyl-CoAs through the catalytic action of long-chain acyl-CoA synthetase (LACS) [12]. The formation of malonyl-CoA, a substrate that is involved in synthesizing very long chain fatty acids (VLCFAs), is catalyzed by acetyl-CoA carboxylase [9]. Cutin monomers of C16 or C18 ω-hydroxylated fatty acids are transported across the plasmalemma by (PM)-localized ATP-binding cassette (ABC) transporters and deposited into the cell wall, where they are assembled in the hydrophobic cuticle layer into a network of linear and branched cutin polymers [13,14].

Our knowledge of the genes underlying cuticular wax biosynthesis, transport, and regulation has expanded greatly, especially in the model organisms *Arabidopsis thaliana* and barley (*Hordeum vulgare* L.). In the past decades, many mutants have been widely used in wax biosynthesis and transport research, revealing the importance of mutants in gene function research for specific biosynthesis and metabolism processes. Wild-type barley leaves are densely covered with small-lobed, crystalline wax plates consisting of predominantly of primary alcohols (C26), while the uppermost internodes, leaf sheaths, and spikes bear a dense array of very long, thin, β-diketone crystalline tubes, predominantly C31 14,16-dione [15]. Many phenotypic changes from glaucous to glossy have been identified by comparing the differences in the mutants with little or no wax loading on the aerial plant organs, which have also been widely used to investigate wax biosynthesis and regulation. These wax-deficient mutants are termed eceriferum (cer) in barley and Arabidopsis, and a total of 85 and 22 cer loci have been identified in barley and Arabidopsis, respectively [8]. In barley, 28 cer genes are specific to the leaf blades, 27 are specific to the uppermost internodes, leaf sheaths, and spikes, and 23 are specific to the spikes only; four of these affected all of the organs [16].

Recently, McAllister et al. [17] confirmed that variation in the wax inducer1 transcription factor *HvWIN1* was caused by cer-x mutants by them controlling the expression of the genes that are involved in the cuticle development in barley. The wax-related gene *BCW1* in barley is located on chromosome 2H, and it has a total length of 15.10 Mb [18]. It is a novel gene regulating cuticular wax biosynthesis and wax crystal formation [18]. *HvCBP20* is a negative regulator of the biosynthesis of waxes at the level of alkane formation and wax transport, causing increased cuticular wax deposition during drought stress [19]. The overall spike wax compositions of 17 spike wax barley mutants have been widely investigated, and it revealed that none of the β-ketoacyl-derived lipids were synthesized in the spike, which was related to Cer-yy regulating gene(s) in the spike for the leaf blade acyl elongase system and repressed the spike acyl and β-ketoacyl elongase systems [20]. Bonus barley spike wax is composed of n-alkanes. In contrast, its cer mutants (cer-a6, cer-e8, cer-n26, and another six mutants) were found that the longer homologs of the n-alkanes greatly reduced. Moreover, its acyl and polyketide elongase systems and the associated reductive and decarbonylative/decarboxylative enzyme systems are defective in synthesizing C32 acyl chains [21]. However, because these mutants are only affected in their total wax load, rather than among their wax components, our current understanding of the precise wax biosynthetic pathway in the spikes is still limited.

In the present study, a spike cuticular wax-deficient barley mutant Cer-GanNong1 (Cer-GN1) was identified from an ethylmethanesulfonate (EMS) mutagenesis population. Compared to the wild-type (WT) plants, the litter cuticular waxes were found on the surface of the spikes, and the total wax load was significantly reduced. We conducted physiological, metabolome, and transcriptome analyses on the Cer-GN1 mutant and WT plants to investigate the changes in crystal morphology and cuticular wax metabolites and identify the major effects of Cer-GN1 mutation on the expression of the other cuticle-related genes. The results of this study provide an ideal mutant for investigating the molecular mechanism of wax crystal self-assembly and a basis for cloning the cuticular wax-related genes from the Cer-GN1 mutant.

## 2. Results

### 2.1. Phenotypic and Physiological Analysis of the Spike Cuticular Wax-Deficient Cer-GN1 Mutant

The plants and cuticular waxes of Cer-GN1 and its WT progenitor cultivar ‘08-DM042’ were observed throughout the growth period. The mutant phenotype of the spikes appeared gradually after the heading stage. Although a white frost wax layer appeared on the surface of the spike of WT plants, the spike surfaces of the Cer-GN1 mutant were glossy. This difference was obvious during the heading to filling stage (Figure 1A,B). We found that the glumes of Cer-GN1 were completely stained by 0.05% TB after 3 h, while the WT glumes had only partial staining of a lighter color when they were compared to Ger-GN1, indicating that Cer-GN1 was defective in its glume cuticle (Figure 1C). The glossy phenotype suggested the absence of cuticular wax, which plays an important role in preventing plant water loss. Therefore, the wax load on the spike surface and water loss rates for the Cer-GN1 and WT plants were investigated. The results showed that the wax load on the spike surfaces of Cer-GN1 was much lower than that of WT, and its content was only one-third of that of the WT plants (Figure 2A). Moreover, the water loss rates for the Cer-GN1 spikes showed a small but significant increase when it was compared to that of WT (*p* < 0.05), especially after 5 h (Figure 2B). The glumes of the spikes from the Cer-GN1 and WT plants in the heading stage were examined by SEM to further investigate the differences in the epicuticular wax crystals of the two plants. We found that the epicuticular wax structures covering the spike surfaces of the Cer-GN1 were different from those of the WT plants (Figure 2). The WT spike surfaces were covered with dense wax crystals with long wax crystal tubes that were woven into the complex network structure (Figure 2C). No obvious wax crystals were deposited on the Cer-GN1 spike surfaces, especially in in the tubule-shaped wax crystals (Figure 2D). These results indicated that Cer-GN1 was a glossy, wax-deficient phenotype of the spike surfaces and it was sensitive to water loss. To determine the relationship between the glossy spike mutant Cer-GN1 and the other epicuticular waxes mutants in the barley, we compared the phenotypic description information that was associated with 82 different epicuticular waxes mutants that were released on the website of International Database for Barley Genes and Barley Genetic Stocks (https://bgs.nordgen.org/index.php?act=kws&kwl=Epicuticular+waxes&kwk=C33%2C34D; accessed on 6 September 2022). According to the comparison of the phenotypic characteristics of the different epicuticular waxes mutants, we found that Cer-GN1 was most similar to the Cer-yy mutant (Stock number: BGS 536), which does not have a surface wax coating on the spike and lacks β-hydroxy-β-diketones and has relatively increased proportions of primary alcohols in its chemical epicuticular wax composition. Further, the gene Cer-yy was located on chromosome 1HS [22].

### 2.2. Genetic Analysis of the Spike Cuticular Wax-Deficient Cer-GN1 Mutant

Because of the great contrast in the wax crystal morphology and density on the spike surfaces of Cer-GN1 compared to WT, we conducted a genetic analysis on the F1 plants and the F2 populations that were developed from the crosses between Cer-GN1 and two barley cultivars (Mei42 and Ganpi 6) to analyze the inheritance behavior of Cer-GN1. The F1 plants were all glaucous on the spike surface, with a glaucousness level that was similar to Mei42 and Ganpi 6. One F2 population of 488 was segregated into 372 glaucous and 116 glossy plants. A second population of 552 F2 individuals contained 417 glaucous and 129 glossy plants. These data were fitted well with a Mendelian segregation ratio of three (glaucous) to one (nonglaucous) (χ^2^ < χ^2^ 0.05, 1 = 3.84; Table 1), indicating that a single recessive gene was responsible for the glossy spike surface phenotype. 

### 2.3. Identification of Differential Metabolites in Spike Glumes between Cer-GN1 and WT

Metabolites were extracted from spike glumes from cuticular wax-deficient Cer-GN1 mutant and WT plants and analyzed using GC-MS metabolomics. According to PCA (Principal Component Analysis) models, two principal components (PCs) were gained from the comparison of the Cer-GN1 mutant and the WT metabolites, indicating that the metabolomes between Cer-GN1 and WT were largely distinguishable (Figure 3A,B). A total of 7675 metabolites were identified, including 696 annotated metabolites (464 from the positive ionization [PI] mode, and 232 from the negative ionization mode [NI]) (Figure 3C,D; Appendix A). The metabolites were mainly classified into four categories: the compounds with biological roles, lipids, phytochemical compounds, and others (Appendix A). Differentially expressed metabolites between the two groups were those with a fold-change cutoff of ≥1.2 or ≤0.83, a *p*-value of <0.05, and a VIP ≥ 1. When it was compared to those of WT, Cer-GN1 had 90 upregulated (65 for PI, and 25 for NI) and 93 downregulated (69 for PI, and 24 for NI) metabolites (Figure 3C,D; Appendix A). A KEGG pathway enrichment analysis demonstrated that the differentially expressed metabolites belonged mainly to metabolic pathways, secondary metabolite biosynthesis, purine metabolism, and cyanoamino acid metabolism (Figure 4A,B; Appendix A).

To determine if the cuticle-related metabolites were affected by the Cer-GN1 mutation, we searched the metabolites belonging to the cutin, suberine, and wax biosynthesis pathways (Figure 4; Appendix A). Interestingly, only 16-hydroxyhexadecanoic acid was enriched in the cutin, suberine, and wax biosynthesis; it was significantly downregulated when it was compared to WT (Figure 4B; Appendix A). 16-hydroxyhexadecanoic acid is one of the predominant C16 and C18 fatty acid-derived cutin monomers [9].

### 2.4. Global Analysis of DEGs in Cer-GN1 and WT Spike Glumes

To investigate the global gene expression profiles of spike cuticular wax regulation in the Cer-GN1 and WT plants, we collected glume samples from the spikes of the Cer-GN1 and WT plants at the grain filling stage. These samples were sequenced using the Illumina Novaseq6000 platform. On average, the results obtained 6,692,758,800 bp, including 6,779,310,900 bp raw reads and 6,591,485,281 bp and 6,670,362,198 bp clean reads after filtering out the low-quality reads from the Cer-GN1 and WT samples, respectively (Appendix A). All of the sequence data for Cer-GN1 and WT have been deposited in the NCBI-SRA database and are accessible in Bio Project PRJNA861888 (BioSample SUB11849747). After filtering out the rRNA-mapped reads, the remaining clean reads were mapped to the barley reference genome for an annotation analysis. From this analysis, 92.96–93.61% of the total clean reads were mapped well (Appendix A). In addition, 7966 novel genes were identified, and the functions of these genes were annotated using BLAST with the public databases (Appendix A).

The DEGs between Cer-GN1 and WT were determined using a threshold with a false discovery rate (FDR) < 0.05 and |log_2_FC| > 1. When it was compared to those of WT, 1575 upregulated and 1074 downregulated genes were identified in Cer-GN1 (Figure 5A and Appendix A). A pathway analysis of the DEGs which was based on the KEGG database was performed. The results showed that the DEGs participated in a wide range of metabolic pathways, including amino sugar and nucleotide sugar metabolism, DNA replication, brassinosteroid biosynthesis, cutin, suberine and wax biosynthesis, and the biosynthesis of secondary metabolites (Figure 5B and Appendix A), indicating that there were differences in the regulation of the spike cuticular wax between WT and Cer-GN1.

### 2.5. Cer-GN1 DEGs Are Associated with Wax Biosynthesis

To determine how wax biosynthesis-related genes were affected by Cer-GN1, we focused on the DEGs that were associated with cutin, suberine, and wax biosynthesis. Only nine DEGs were involved in cutin, suberine, and wax biosynthesis, including two novel genes (*HvMSTRG.29184* and *HvMSTRG.29185*) (Figure 5C and Appendix A). These two novel genes encode CYP704B1 (long-chain fatty acid omega-monooxygenase), and both of them are downregulated in Cer-GN1 (Figure 5C; Appendix A). The gene *HvMSTRG.29184* was located on chromosome 4H with a 2377 bp (563,743,184–563,745,561 bp) region containing four exons (563,743,184–563,744,280 bp, 563,744,376–563,744,672 bp, 563,744,772–563,744,987 bp, and 563,745,075–563,745,561 bp). Similarly, the gene *HvMSTRG.29185* was located on chromosome 4H with a 4088 bp (563,743,203–563,747,291 bp) region containing two exons (563,743,203–563,743,626 bp and 563,747,203–563,747,291 bp). CYP704B1 regulates the conversion of C16 palmitic acid to 16-hydroxypalmitate (16-hydroxyhexadecanoic acid). This result was consistent with the 16-hydroxyhexadecanoic acid deficiency which was observed for Cer-GN1 by a metabolomics analysis. Moreover, during the VLCFA synthesis, upregulated aldehyde decarbonylase could promote the synthesis of a long-chain aldehyde to a long-chain alkane, while the expression levels of the alcohol-forming fatty acyl-CoA reductase (FAR) were both upregulated and downregulated during the conversion of a long-chain acyl-CoA to a long-chain primary alcohol, respectively. Like the FAR genes, the expression of peroxygenase (PXG) was both upregulated and downregulated during the conversion of C18 oleic acid to 9,10-epoxystearate or 18-hydroxyoleate to 9,10-epoxy-18-hydroxyoleate (Figure 5C; Appendix A), respectively. It should be noted that the conversion of C16 palmitic acid to 16-hydroxypalmitate begins the VLCFA synthesis for wax. Therefore, it is reasonable to speculate that *HvMSTRG.29184* and HvMSTRG.29185 are candidate genes that are involved in regulating wax synthesis in Cer-GN1. The gene sequences of *HvMSTRG.29184* and *HvMSTRG.29185* have been deposited in GenBank under the accession nos. MW999597 and MW999598, respectively.

### 2.6. Verification of HvMSTRG.29184 and HvMSTRG.29185 Gene Expression by qRT-PCR

To further validate the results of this study, the expression levels of *HvMSTRG.29184* and *HvMSTRG.29185* in Cer-GN1 and WT spike glume samples were analyzed at different growth stages 1, 4, 8, 12, and 16 days after the heading (DAH) stage by qRT-PCR. *HvMSTRG.29184* and HvMSTRG.29185 were significantly downregulated in Cer-GN1 when they were compared to their levels in WT (Figure 6), especially at 12 and 16 DAH. These results were consistent with the RNA-seq data, indicating that our analysis was reliable. These data suggest that *HvMSTRG.29184* and HvMSTRG.29185 were candidate genes that could play roles in regulating cuticular wax synthesis.

## 3. Discussion

Because the cuticle plays a significant role in plant stress resistance, cuticular waxes have been widely studied using genetic and biochemical approaches. Generally, wax-deficient mutants are termed eceriferum (cer), typically exhibiting a glossy phenotype. They are intensively used for studying wax biosynthesis and transport processes, and cer is used as the symbol for the corresponding gene loci [23,24]. Since the initial seventeen mutations resulting in a phenotypic deficiency in the epicuticular wax on barley spikes were identified [20], more than 90 cer have been found in barley. However, our present knowledge of the wax biosynthetic pathway and the precise regulation of the particular enzymes remains fragmentary. Some cer genes have been localized, cloned, and confirmed to play a role in barley wax synthesis [4,25,26]. In the present study, a barley cuticular wax-deficient mutant (Cer-GN1) was identified from an EMS mutagenesis population, and it was shown to have a glossy spike phenotype with there being no other obvious differences when it was compared to its WT progenitor. The characterization of this mutant confirmed that it was defective in the glume cuticle. Indeed, the total cuticular wax load of the spikes was significantly decreased in Cer-GN1, and its water loss rate showed a corresponding increase. Furthermore, SEM confirmed that the tubule-shaped crystals that are normally deposited on the glume surfaces were absent in Cer-GN1.

Tubule crystals are known to be predominantly β-diketone or secondary alcohols [27,28]. Changes in the wax crystal structure in barley have been widely identified using the mutants covering the tissues of the leaf, stem, leaf sheaths, leaf blades, and spikes. The *cer-j.59* mutant has single, irregularly scattered wax bodies in its leaves due to it having a large decrease in primary alcohols and an increase in the compound class of esters [29]. The stem and sheath surfaces of the *bcw1* mutant are glossy, showing only the little platelet-shaped waxy crystals that are deposited on the tissue surface [18]. The *gsh4.l* mutant has a striking glossy appearance on its spikes, leaf sheaths, and leaf blades. This mutation alters specific cuticle structures by causing a loss of β-diketones, VLC alkanes, and VLCFAs. The *cer.i16* mutant has shorter tubes which are peculiar to spikes, which have about a 36% reduction in the number of β-diketones [16]. Changes in the composition and total wax load of these mutants are closely related to the regulation of the wax biosynthetic pathway.

We analyzed the chemical composition of the glumes of the Cer-GN1 mutant and the WT plants using metabolomics to investigate which compositional changes in the cuticle waxes caused the loss of glaucous wax in Cer-GN1. We identified significant differences in the metabolites between the Cer-GN1 mutant and the WT plants. Forty metabolites belonged to the lipid metabolic pathways, and only 16-hydroxyhexadecanoic acid was downregulated in the Cer-GN1 mutant when it was compared to the level of it in WT. This result is different from the chemical epicuticular wax composition of the glossy spike mutant Cer-yy, which lacks β-hydroxy-β-diketones and has relatively increased proportions of primary alcohols, although both of the mutants Cer-GN1 and Cer-yy have similar phenotypes (https://bgs.nordgen.org/index.php?pg=bgs_show&docid=253; accessed on 6 September 2022). Combining the glossy spike phenotype and the chemical epicuticular wax composition characteristics, the glossy spike mutant Cer-GN1 is a phenotype that has not been characterized at the present time. The 16-hydroxyhexadecanoic acid is enriched in the cutin, suberine, and wax biosynthesis pathways and is one of the predominant C16 and C18 fatty acid-derived cutin monomers [9]. Cutin consists of a large amount of polyesters whose monomer composition is mainly composed of C16 and C18 hydroxylated fatty acids and their derivatives and phenolic compounds; its biosynthesis starts with the formation of cutin monomers [13,14]. Cuticular wax formation starts with the de novo biosynthesis of C16 and C18 acyl-CoAs in the plastids of epidermal cells [2]. The C16 and C18 acyl-CoAs can be elongated into very-long-chain fatty acids (VLCFAs) (C20 to C34) by the fatty acid elongase complex that includes β-ketoacyl-CoA synthetase (KCS), β-ketoacyl-CoA reductase (KCR), β-hydroxyacyl-CoA dehydratase (HCD), and enoloyl-CoA reductase (ECR) [30]. Finally, most of these VLCFAs are used for cuticular wax production and they are converted into different waxy compounds by acyl reduction and decarbonization [31]. Therefore, it is reasonable to presume that inhibiting the 16-hydroxyhexadecanoic acid synthesis could reduce the subsequent cutin and wax synthesis, causing the loss of glaucous wax from the spikes of Cer-GN1 barley plants.

Based on the particularity of the wax composition of the glossy spike mutant Cer-GN1, we also performed a comparative transcriptomic analysis on the glumes from the Cer-GN1 mutant and the WT plants at the grain filling stage to identify the major effects of the Cer-GN1 mutation on the expression of the genes that are involved in cuticular wax synthesis. Our results showed that nine DEGs were involved in cutin and wax synthesis and its regulation. In particular, we identified two novel genes (*HvMSTRG.29184* and *HvMSTRG.29185*) that are located on chromosome 4H which were downregulated in the Cer-GN1 mutant, which was also different from the result of the localization of the gene *Cer-yy* on chromosome 1HS [22]. The genes *HvMSTRG.29184* and *HvMSTRG.29185* encode the long-chain fatty acid omega-monooxygenase CYP704B1, which regulates the conversion of C16 palmitic acid to 16-hydroxyhexadecanoic acid. The downregulation of *HvMSTRG.29184* and *HvMSTRG.29185* inhibited the synthesis of 16-hydroxyhexadecanoic acid in the waxless plants after the heading phase, which is consistent with our metabolomics data showing the reduced synthesis of 16-hydroxyhexadecanoic acid in the Cer-GN1 mutant. Many genes that are involved in wax synthesis and regulation have been previously identified in plants. *HvKCS1* plays a major role in the total acyl chain elongation for wax biosynthesis due to its substrate specificity for C16–C20 [4]. *HvFDH1;1*, which is related to *Arabidopsis* Fiddlehead (FDH), is likely involved in VLCFA synthesis [32]. *LGF1* is necessary for the abundance of epicuticular wax platelets by regulating the C30 primary alcohol synthesis in rice [33]. *BnWIN1* overexpression enhances the wax accumulation in *Brassica napus* by increasing the C29-alkane, C31-alkane, C28-alcohol, and C29-alcohol content [34]. However, only a few genes encoding CYP704B1 have been reported. In *Arabidopsis*, At1g69500 encodes CYP704B1 and catalyzes the ω-hydroxylation of long-chain fatty acids [35]. Increased cutin monomer levels in the siliques of *Panax ginseng* cause *PgCYP704B1* overexpression [36]. Therefore, we can speculate that *HvMSTRG.29184* and *HvMSTRG.29185* regulate the conversion of C16 palmitic acid to 16-hydroxyhexadecanoic acid in barley via their CYP704B1 function, reducing the synthesis of subsequent cuticular wax components, which ultimately results in the loss of glaucous wax from the spikes of Cer-GN1 mutant plants (Figure 7).

## 4. Materials and Methods

### 4.1. Plant Material and Growth Conditions

The selfing progeny of the spike cuticular wax-deficient Cer-GN1 mutant generated by EMS mutagenesis and its progenitor wild-type (WT) cultivar ‘08-DM042’ were obtained from the College of Agronomy, Gansu Agricultural University [37]. All of the experimental materials were subjected to standard cultivation at the Gansu Agricultural University greenhouse. Briefly, plants were cultivated on acid soil and uniformly sown, and they were fertilized and watered in greenhouse-controlled conditions under a 16 h light/8 h dark cycle with relative humidity 40–45%. Cer-GN1 was crossed with two barley cultivars, Mei 42 and Ganpi 6, to construct the F1 and F2 populations. The phenotypes of the F1 and F2 progenies were observed from heading to ripening, and the numbers of normal and mutant phenotypic plants in two segregating populations were recorded. Because of the obvious phenotypic differences in the cuticular wax of the spike between the Cer-GN1 mutant and WT plants, they were used to compare the cuticular wax metabolites and transcriptome analyses. Seeds from the Cer-GN1 and WT plants were sown in 25 cm pots with commercial compost, and the plants were grown under a cycle of 16 h light at 22 °C/8 h dark at 12 °C. The relative air humidity was 60%.

### 4.2. Spike Sampling and Analysis

Spikes were cut from fresh Cer-GN1 mutant and WT plants at the heading stage and divided into four groups. One group was used for spike cuticular wax observations, and a second group was used to determine the water loss rate. The final two groups were used for metabolite and RNA extraction. For the spike cuticular wax observations, the spikes were immersed in 0.05% (wt/vol) toluidine blue (TB) for 3 h, and then, they were washed under water to remove unbound dye before scanning [38]. Segments from the adaxial surface of the glume were attached with double-sided tape to the stubs to analyze the epicuticular wax crystallization patterns using a JEOL JSM-5600LV scanning electron microscope (SEM) at 20 kV. The water loss rate of the spikes was determined by the natural drying method. Cer-GN1 and WT spikes at the heading stage were collected and dried naturally in the laboratory at room temperature. The weight of the spikes were measured at 14 time points. The water loss of the spikes is expressed as a percentage of the initial weight h–1. The experiment was repeated six times. For metabolite extraction, six independent replicate samples were harvested. For each replicate, 20 spikes were randomly collected from four individual plants. For RNA extraction, three independent replicate samples were harvested in a similar manner.

### 4.3. Metabolite Extraction, Detection, and Analysis

Freeze-dried samples were ground to power using a grinder for 1.5 min at 30 Hz. The powder (100 mg) was dissolved in 1000 μL methanol (−20 °C), vortexed for 1 min, and then centrifuged at 12,000 rpm for 10 min at 4 °C. The supernatant was transferred to a 2 mL centrifuge tube and vacuum-dried. The extracts were redissolved in a 2-chlorobenzalanine (4 ppm) 80% methanol solution, and the supernatant was filtered through a 0.22 μm membrane to obtain the samples for UPLC-MS analysis.

Untargeted metabolomics analysis was performed at Gene Denovo Biotechnology Co. (Guangzhou, China). Chromatographic separation was accomplished using a Thermo Ultimate 3000 system that was equipped with an ACQUITY UPLC^®^ HSS T3 (150 × 2.1 mm, 1.8 μm, Waters, Milford, MA, USA) column. Gradient elution of the analytes was carried out with 0.1% formic acid in water (A) and 0.1% formic acid in acetonitrile (B) at a flow rate of 0.30 mL/min. An increasing linear gradient of solvent B (*v*/*v*) was used as follows: 0–1 min, 2% B; 1–9 min, 2–50% B; 9–12 min, 50–98% B; 12–13.5 min, 98% B; 13.5–14 min, 98–2% B; 14–20 min, 2% B-positive model (14–17 min, 2% B-negative model). The ESI-MSn experiments were executed using the Thermo Q Exactive mass spectrometer with a spray voltage of 3.8 kV and −2.5 kV in the positive and negative modes, respectively. The sheath and auxiliary gases were set at 30 and 10 arbitrary units, respectively. The capillary temperature was 325 °C. The analyzer scanned over a mass range of *m*/*z* 81–1000 for the full scan at a mass resolution of 70,000. Data-dependent acquisition (DDA) MS/MS experiments were performed by an HCD scan. The normalized collision energy was 30 eV. Unnecessary information in the MS/MS spectra was removed by dynamic exclusion. A fold-change cutoff of ≥1.2 or ≤0.83, *p*-value < 0.05, and variable importance in projection (VIP) ≥ 1 were used to identify differentially expressed metabolites between the two groups. Metabolite annotations and pathway enrichment analysis were based on the Kyoto Encyclopedia of Genes and Genomes (KEGG) database.

### 4.4. RNA Extraction, Library Construction, and Sequencing

Total RNA was extracted using Trizol reagent (Invitrogen, Carlsbad, CA, USA), according to the manufacturer’s protocol. RNA quality was assessed using an Agilent 2100 Bioanalyzer (Agilent Technologies, Palo Alto, CA, USA) and checked using RNase-free agarose gel electrophoresis. Eukaryotic mRNA was enriched by Oligo(dT) beads, while prokaryotic mRNA was enriched by removing rRNA using the Ribo-ZeroTM Magnetic Kit (Epicentre, Madison, WI, USA). The enriched mRNA was fragmented into short fragments using fragmentation buffer, and it was reverse transcribed into cDNA with random primers. Second-strand cDNA was synthesized using DNA polymerase I, RNase H, dNTPs, and buffer. The cDNA fragments were purified using the QIAquick PCR extraction kit (Qiagen, Venlo, The Netherlands) and end-repaired. Poly(A) was added to the cDNA fragments, which were then ligated to Illumina sequencing adapters. The cDNA libraries (volume 60 μL; concentration 7.925–9.894 ng/μL) with three biological replicates were prepared using the RNA-Seq. The ligation products were size-selected by agarose gel electrophoresis, PCR-amplified, and sequenced using the Illumina HiSeq2500 by Gene Denovo Biotechnology Co. (Guangzhou, China). 

### 4.5. Bioinformatics Analysis

Reads that were obtained from sequencing machines included raw reads containing adapters or low-quality bases, which would affect the subsequent assembly and analysis. Thus, the reads were further filtered by fastp version 0.18.0 to get high-quality clean reads [39]. The filtering parameters included: (1) removing the reads containing the adapters; (2) removing reads containing more than 10% unknown nucleotides (N); (3) removing low-quality reads containing more than 50% low-quality bases (Q-value ≤ 20). The short reads alignment tool Bowtie2 version 2.2.8 was used for mapping reads to the ribosome RNA (rRNA) database [40]. The rRNA-mapped reads were removed. The remaining clean reads were used in the assembly and gene abundance calculation. The paired-end clean reads were mapped to the barley reference genome of barley cv. Morex (MorexV3) [41] using HISAT2. 2.4 with “-rna-strandness RF” and other parameters set as the default [42]. The mapped reads of each sample were assembled using StringTie v1.3.1 with a reference-based approach [43]. An FPKM (fragment per kilobase of transcript per million mapped reads) value was calculated for each transcription region to quantify its expression abundance and variations using StringTie software [44]. Genes that were found in this sequencing results, but not those that were included in the reference genome, were defined as novel genes and represented by HvMSTRG. Differential RNA expression analysis was performed using DESeq2 software [45] between the Cer-GN1 mutant and WT groups. Genes with a false discovery rate (FDR) below 0.05 and an absolute fold-change ≥ 2 were considered differentially expressed genes (DEGs). KEGG enrichment analysis was performed to analyze the biological functions and metabolic pathways that were represented by the DEGs.

### 4.6. Quantitative RT-PCR Analysis

Quantitative RT-PCR (qRT-PCR) analysis was performed to verify the RNA-Seq results based on candidate genes (*HvMSTRG.29184* and *HvMSTRG.29185*) that were involved in regulating wax synthesis in Ger-GN1. The primers for Ger-GN1*HvMSTRG.29184* were 5′-TCTCGGTCGGCTTGATTTA-3′ (forward primer) and 5′-GGTACGATGGTAGCAGGGAC-3′ (reverse primer). The primers for *HvMSTRG.29185* were 5′-TACGATGGTAGCAGGGAC-3′ (forward primer) and 5′-CGTAGGCAGAGTGGACAG-3′ (reverse primer). The length of PCR products for *HvMSTRG.29184* and *HvMSTRG.29185* were 123 bp and 121 bp, respectively. The spike RNA samples that were used for qRT-PCR were collected randomly from three individual plants 1, 4, 8, 12, and 16 days after heading (DAH). The RNA extraction and cDNA synthesis were performed as described for RNA-seq. qRT-PCR was performed using SYBR PremixDimerEraser™ (Takara Bio, Otsu, Japan) according to the manufacturer’s specifications. The PCR reaction was performed using the QuantStudio 5 Real-Time PCR System (Applied Biosystems, Foster City, CA, USA) at 95 °C for 3 min, followed by 40 cycles of 95 °C for 10 s, 60 °C for 60 s, and 72 °C for 30 s. Barley *ACTIN* (AY145451) was used as a control to normalize the amount of gene-specific RT-PCR products (forward primer: 5′-GCCGTGCTTTCCCTCTATG-3′; reverse primer 5′-GCTTCTCCTTGATGTCCCTTA-3′; the length of PCR product was 235 bp) [46]. The relative expression levels of the selected genes were calculated with the 2^−ΔΔCt^ method [47].

## 5. Conclusions

Overall, these results indicated that mutation of Cer-GN1 leads to the loss of glaucous wax in the spikes by reducing the amount of tubule-shaped crystals that were depositing on the spikes. Comparative metabolomics and transcriptomics between the Cer-GN1 mutant and WT plants revealed that inhibition of the synthesis of 16-hydroxyhexadecanoic acid, one of the predominant C16 and C18 fatty acid-derived cutin monomers, played a major role in reducing the cuticular wax levels in the spikes of Cer-GN1. This process was regulated by CYP704B1, which was encoded by two novel genes that are located on chromosome 4H, *HvMSTRG.29184* and *HvMSTRG.29185*. It remains to be investigated whether *HvMSTRG.29184* and *HvMSTRG.29185* truly have CYP704B1 activity that regulates the conversion of C16 palmitic acid to 16-hydroxyhexadecanoic acid in barley. Moreover, our results provide an ideal mutant for investigating the molecular mechanism of wax cuticular biosynthesis and a basis for cloning the cuticular wax-related genes from the Cer-GN1 mutant.

## Figures and Tables

**Figure 1 ijms-23-13025-f001:**
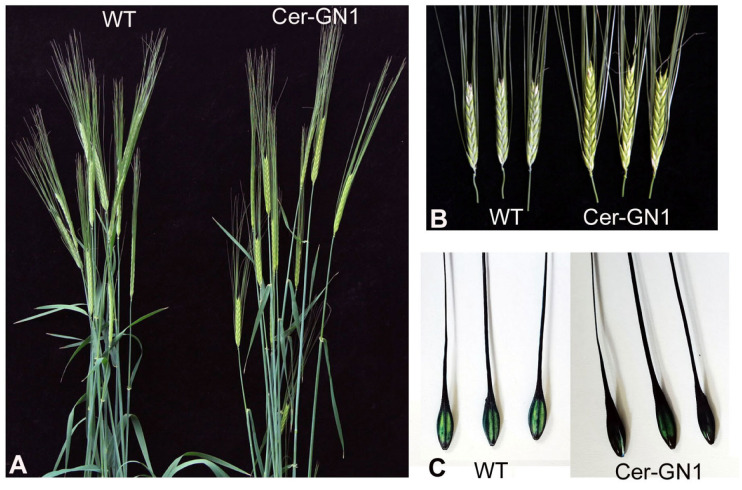
The phenotypes of Cer-GN1 mutant and WT plants. Plants at the grain filling stage (**A**). The cuticular wax phenotype on the spikes (**B**). TB staining of the glume surface (**C**).

**Figure 2 ijms-23-13025-f002:**
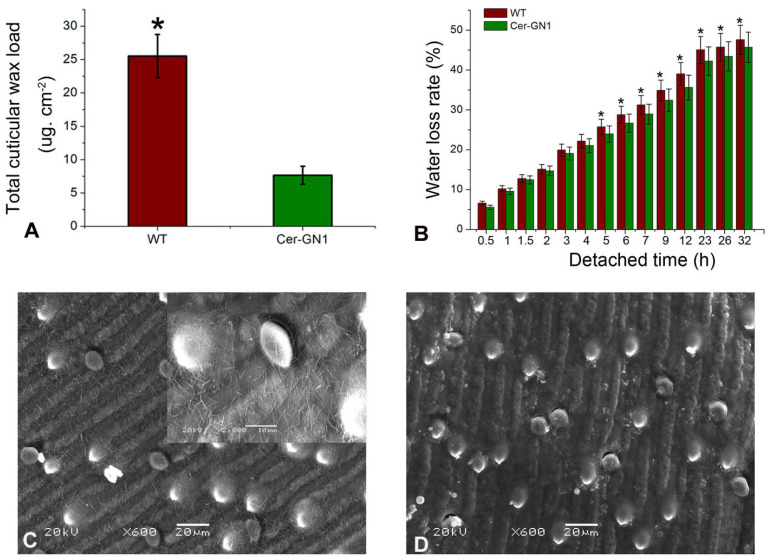
Cuticular wax variations of Cer-GN1 mutant and WT plants. The total cuticular wax load on the glume surface (**A**). Water loss rate (**B**). SEM of wax crystals on the glume surfaces of Cer-GN1 (**C**) and WT (**D**). Each value represents the mean of three replicates ± SD (*n* = 3). * *p* < 0.05 by the Student’s *t*-test.

**Figure 3 ijms-23-13025-f003:**
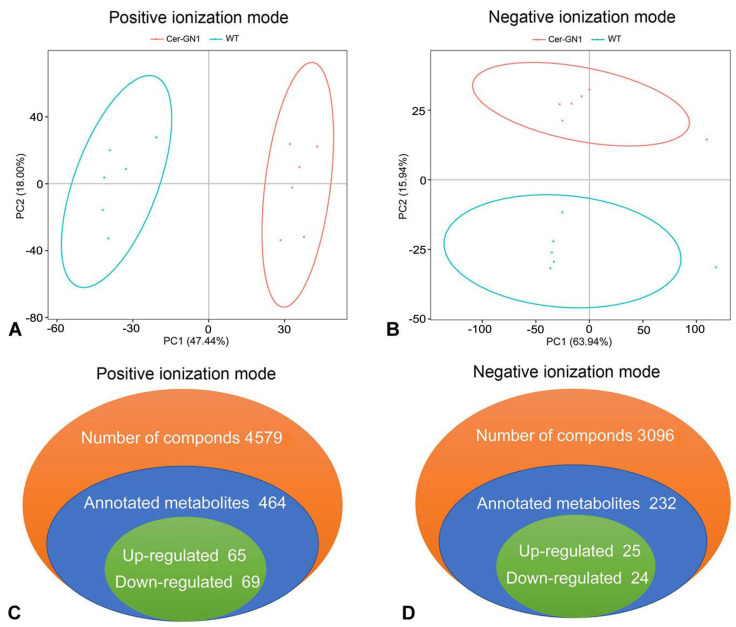
Metabolite properties of the glumes in Cer-GN1 mutant and WT plants. Overall score plots of the PCA model in the positive (**A**) and negative (**B**) ionization modes, respectively. Summary of the metabolomics quantified in the positive (**C**) and negative (**D**) ionization modes, respectively.

**Figure 4 ijms-23-13025-f004:**
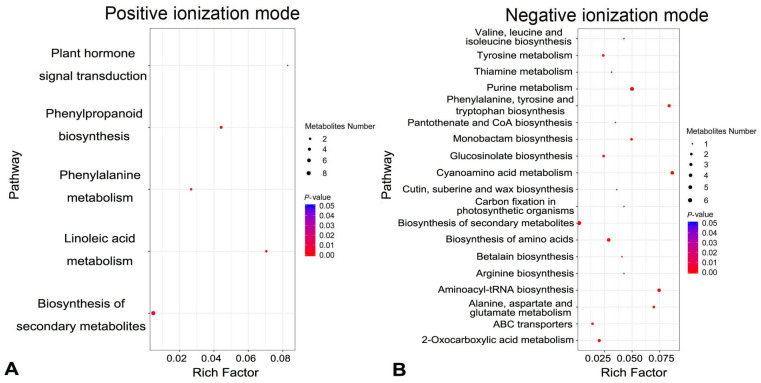
KEGG pathway enrichment analysis for differentially expressed metabolites in the positive (**A**) and negative (**B**) ionization modes, respectively.

**Figure 5 ijms-23-13025-f005:**
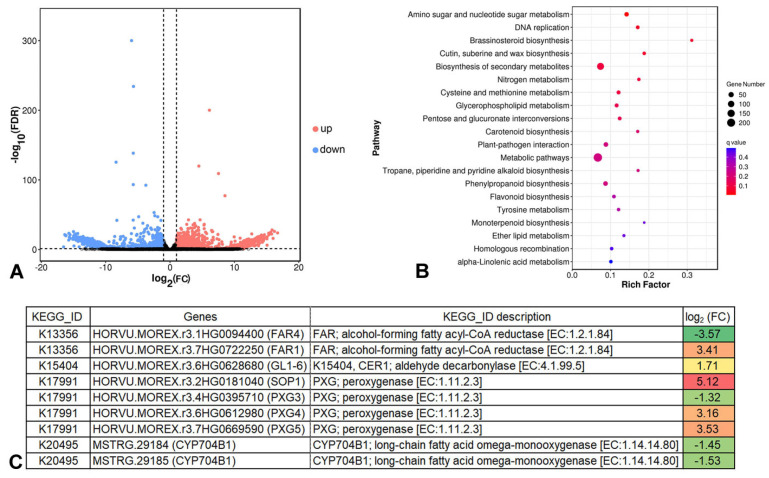
Properties of the differentially expressed genes (DEGs) in the glumes of Cer-GN1 mutant and WT plants identified by RNA-Seq. Expression levels of the DEGs in the Cer-GN1 and WT libraries (**A**). KEGG pathway enrichment analysis of the DEGs (**B**). Gene information belonging to the metabolic pathway of cutin, suberine, and wax biosynthesis (**C**).

**Figure 6 ijms-23-13025-f006:**
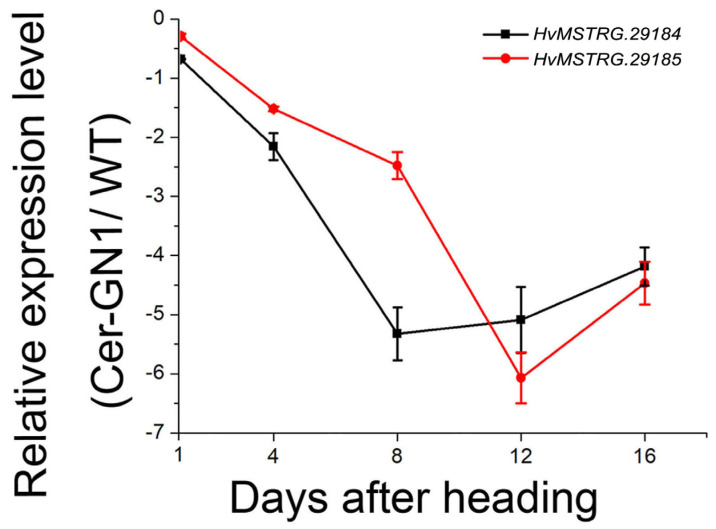
qRT-PCR validation of candidate genes *HvMSTRG.29184* and *HvMSTRG.29185* regulating wax synthesis in the Cer-GN1 mutant. The cuticular wax phenotype of the spikes of Cer-GN1 mutant and WT plants at 1, 4, 8, 12, and 16 days after heading (DAH). The scale bars represent the standard deviation of three biological replicates. Relative RNA expression levels for HvMSTRG.29184 and HvMSTRG.29185 were determined using mutant and WT gene-specific primers.

**Figure 7 ijms-23-13025-f007:**
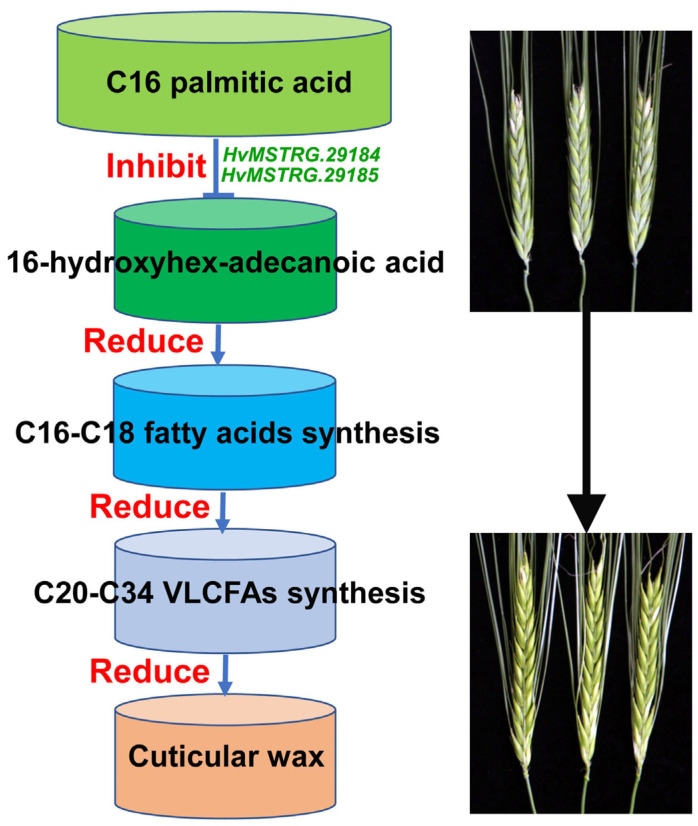
Simplified model for the function of novel genes *HvMSTRG.29184* and *HvMSTRG.29185* in regulating wax synthesis in the Cer-GN1 mutant. Downregulation of *HvMSTRG.29184* and *HvMSTRG.29185* would lead to the reduced accumulation of the predominant C16 and C18 fatty acid-derived 16-hydroxyhexadecanoic acid cutin monomer by inhibiting the conversion of the C16 of palmitic acid to 16-hydroxyhexadecanoic acid, which causes C16–C18 fatty acid synthesis, VLCFA synthesis and cuticular wax are decreased in the Cer-GN1 mutant.

**Table 1 ijms-23-13025-t001:** Characteristics of the waxy spike phenotype of F1 and F2 genetic populations.

Crossing Combination	F1 Plants	F2 Populations				
		Total Plants	Glaucous Plants	Glossy Plants	χ^2^ 3:1	χ^2^ 0.05
Cer-GN1 × Mei 42	Glaucous	488	372	116	0.3307	3.84
Cer-GN1 × Ganpi 6	Glaucous	552	417	129	0.0604	3.84

## Data Availability

All the sequence data for Cer-GN1 and WT have been deposited in the NCBI-SRA database and are accessible in Bio Project PRJNA861888 (BioSample SAMN29931758-SAMN29931763). The gene sequences of *HvMSTRG.29184* and *HvMSTRG.29185* have been deposited in GenBank under the accession no. OP179628 and OP179629, respectively.

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
