# Peer review of "Characterization of Glossy Spike Mutants and Identification of Candidate Genes Regulating Cuticular Wax Synthesis in Barley (Hordeum vulgare L.)"

_ijms, 2022, doi:10.3390/ijms232113025_

Round 1
Reviewer 1 Report (Previous Reviewer 2)
Dear Author,
your manuscript entitled “Characterization of glossy spike mutants and identification of candidate genes regulating cuticular wax synthesis in barley (Hordeum vulgare L.)” is well written, and the research is organized consistently with the objectives. Results have been properly reported and discussed.
Few minor revisions are required.
Figure 4 is not readable. The conclusion should be rewritten especially the last sentence to point out the novelty of your results.
Sincerely
Author Response
Dear reviewer,
We thank you for your encouraging comments on our manuscript (ID: ijms-1936869). After carefully considering all of the comments, we have performed the point-by-point revisions. Please find our responses to these comments/suggestions listed below.
Sincerely yours,
Huajun Wang
[Comment] Figure 4 is not readable. The conclusion should be rewritten especially the last sentence to point out the novelty of your results.
[Response] Thank you for this very important comment. Based on your advice, we have modified the related content in the manuscript. For Fig.4, we have put Fig. 4A and B in Supplementary Fig. S1, redraw 4C and 4D, modified the font size, and finally formed new Fig. 4A and B (line 196-199). In addition, similar to Fig. 4, we also modified the font size in Fig. 3 and Fig. 6 (line 185-187; line 264-265).
For the conclusion part, some sentences have been re-written and added (line 482-493). Overall, these results indicated that mutation of Cer-GN1 leads to the loss of glaucous wax in the spikes by reducing the amount of tubule-shaped crystals depositing on the spikes. Comparative metabolomics and transcriptomics between the Cer-GN1 mutant and WT plants revealed that inhibition of the synthesis of 16-hydroxyhexadecanoic acid, one of the predominant C16 and C18 fatty acid-derived cutin monomers, played a major role in reducing the cuticular wax levels in the spikes of Cer-GN1. This process was regulated by CYP704B1, encoded by two novel genes located on chromosome 4H, HvMSTRG.29184 and HvMSTRG.29185. It remains to be investigated whether HvMSTRG.29184 and HvMSTRG.29185 truly have CYP704B1 activity that regulates the conversion of C16 palmitic acid to 16-hydroxyhexadecanoic acid in barley. Moreover, our results provided an ideal mutant for investigating the molecular mechanism of wax cuticular biosynthesis and a basis for cloning cuticular wax-related genes from the Cer-GN1 mutant.

Reviewer 2 Report (Previous Reviewer 3)
I feel this article does not meet the standard of current barley genetics. I decline any further commenting.
Author Response
Dear reviewer,
We thank you for your comments on our manuscript (ID: ijms-1936869). After carefully considering all of the comments, we have performed the point-by-point revisions. Please find our responses to these comments/suggestions listed below.
Sincerely yours,
Huajun Wang
Responses to Reviewer 1:
[Comment] Figure 4 is not readable. The conclusion should be rewritten especially the last sentence to point out the novelty of your results.
[Response] Thank you for this very important comment. Based on your advice, we have modified the related content in the manuscript. For Fig.4, we have put Fig. 4A and B in Supplementary Fig. S1, redraw 4C and 4D, modified the font size, and finally formed new Fig. 4A and B (line 196-199). In addition, similar to Fig. 4, we also modified the font size in Fig. 3 and Fig. 6 (line 185-187; line 264-265).
For the conclusion part, some sentences have been re-written and added (line 482-493). Overall, these results indicated that mutation of Cer-GN1 leads to the loss of glaucous wax in the spikes by reducing the amount of tubule-shaped crystals depositing on the spikes. Comparative metabolomics and transcriptomics between the Cer-GN1 mutant and WT plants revealed that inhibition of the synthesis of 16-hydroxyhexadecanoic acid, one of the predominant C16 and C18 fatty acid-derived cutin monomers, played a major role in reducing the cuticular wax levels in the spikes of Cer-GN1. This process was regulated by CYP704B1, encoded by two novel genes located on chromosome 4H, HvMSTRG.29184 and HvMSTRG.29185. It remains to be investigated whether HvMSTRG.29184 and HvMSTRG.29185 truly have CYP704B1 activity that regulates the conversion of C16 palmitic acid to 16-hydroxyhexadecanoic acid in barley. Moreover, our results provided an ideal mutant for investigating the molecular mechanism of wax cuticular biosynthesis and a basis for cloning cuticular wax-related genes from the Cer-GN1 mutant.
Responses to Reviewer 3:
[Comment 1] Sentence in lines 19-20 should be rewritten
[Response 1] Thank you for the advice. The sentence has been re-written (line 19-20). Besides, we found that there were some language problems throughout the paper. We have corrected these mistakes in the revised manuscript (marked in red)
[Comment 2] Line 99- provide the meaning of the GN1 abbreviation
[Response 2] Thank you for the suggestion. We have added the related content in
the manuscript (line 98-99). The GN1 abbreviation represent GanNong1. Here, GanNong is the abbreviation of Chinese name of our university (Gansu Nongye Daxue), Gansu Agricultural University. So, in order to better distinguish the spike cuticular wax-deficient barley mutant identified from an ethylmethanesulfonate (EMS) mutagenesis population, we used Cer-GanNong1 (Cer-GN1) to represent the name of this mutant material.
[Comment 3] Section 4.4 provide the concentration and volume of the library
[Response 3] Thank you for the advice. We have added the related content in
the manuscript (line 435-436).
The cDNA libraries (volume 15 ul; concentration 7.925-9.894 ng/ul) with three biological replicates were prepared using the RNA-Seq.
[Comment 4] Section 4.6 Provide the length of PCR products (both tested and reference actin gene), name and manufacturer of qPCR equipment.
[Response 4] Thank you for the suggestion. We have added the related content in
the manuscript.
Line 469-470: The length of PCR products for HvMSTRG.29184 and HvMSTRG.29185 were 123 bp and 121 bp, respectively.
Line 474-479: The PCR reaction was performed in the QuantStudio 5 Real-Time PCR System (Applied Biosystems, Foster City, CA, USA) at 95℃ for 3 min, followed by 40 cycles of 95℃ for 10 s, 60℃ for 60 s, and 72℃ for 30 s. Barley ACTIN (AY145451) was used as a control to nor-malize the amount of gene-specific RT-PCR products (forward primer: 5'-GCCGTGCTTTCCCTCTATG-3'; reverse primer 5'-GCTTCTCCTTGATGTCCCTTA-3'; the length of PCR product was 235 bp) [46].
Reviewer 3 Report (New Reviewer)
Authors performed analysis of barley mutants indicating reduced cuticular wax biosynthesis and identified two genes that may be responsible for this phenotype. Research is of good quality, well planned and performed. Obtained results support presented conclusions. Minor corrections presented below should be introduced before the publication.
Sentence in lines 19-20 should be rewritten
Line 99- provide the meaning of the GN1 abbreviation
Section 4.4 provide the concentration and volume of the library
Section 4.6 Provide the length of PCR products (both tested and reference actin gene), name and manufacturer of qPCR equipment.
Author Response
Dear reviewer,
We thank you for your encouraging comments on our manuscript (ID: ijms-1936869). After carefully considering all of the comments, we have performed the point-by-point revisions. Please find our responses to these comments/suggestions listed below.
Sincerely yours,
Huajun Wang
[Comment 1] Sentence in lines 19-20 should be rewritten
[Response 1] Thank you for the advice. The sentence has been re-written (line 19-20). Besides, we found that there were some language problems throughout the paper. We have corrected these mistakes in the revised manuscript (marked in red).
[Comment 2] Line 99- provide the meaning of the GN1 abbreviation
[Response 2] Thank you for the suggestion. We have added the related content in
the manuscript (line 98-99). The GN1 abbreviation represent GanNong1. Here, GanNong is the abbreviation of Chinese name of our university (Gansu Nongye Daxue), Gansu Agricultural University. So, in order to better distinguish the spike cuticular wax-deficient barley mutant identified from an ethylmethanesulfonate (EMS) mutagenesis population, we used Cer-GanNong1 (Cer-GN1) to represent the name of this mutant material.
[Comment 3] Section 4.4 provide the concentration and volume of the library
[Response 3] Thank you for the advice. We have added the related content in
the manuscript (line 435-436).
The cDNA libraries (volume 15 ul; concentration 7.925-9.894 ng/ul) with three biological replicates were prepared using the RNA-Seq.
[Comment 4] Section 4.6 Provide the length of PCR products (both tested and reference actin gene), name and manufacturer of qPCR equipment.
[Response 4] Thank you for the suggestion. We have added the related content in
the manuscript.
Line 469-470: The length of PCR products for HvMSTRG.29184 and HvMSTRG.29185 were 123 bp and 121 bp, respectively.
Line 474-479: The PCR reaction was performed in the QuantStudio 5 Real-Time PCR System (Applied Biosystems, Foster City, CA, USA) at 95℃ for 3 min, followed by 40 cycles of 95℃ for 10 s, 60℃ for 60 s, and 72℃ for 30 s. Barley ACTIN (AY145451) was used as a control to nor-malize the amount of gene-specific RT-PCR products (forward primer: 5'-GCCGTGCTTTCCCTCTATG-3'; reverse primer 5'-GCTTCTCCTTGATGTCCCTTA-3'; the length of PCR product was 235 bp) [46].

This manuscript is a resubmission of an earlier submission. The following is a list of the peer review reports and author responses from that submission.
Round 1
Reviewer 1 Report
This study reported the physical characterization of a barley glossy spike mutant Cer-GN1, which showed ca. 30% of the cuticular wax in wild type. A genetic analysis suggested this trait is controlled by a single recessive nuclear gene. Using metabolomics and transcriptomics analysis, the impact of cuticular wax composition and waxy regulatory genes on the loss of glaucous wax in the spikes of Cer-GN1 were investigated. 16-hydroxyhexadecanoic acid, one of the predominant C16 and C18 fatty acid-derived cutin monomers, was significantly down-regulated in Cer-GN1 compared to WT.
I have few comments rather minors regarding to this manuscript.
The barley genome reference Morex v3 (Mascher et al. 2021) is well accepted in barley research community. I would suggest to modify the gene IDs MSTRG.29184 and MSTRG.29185 in main text and legends.
Figure 6, the pictures (A-E) are pool for making interpretation. These panels are dispensable. Better to remove them.
The reference genome for reads mapping is not given in Methods.
Author Response
Dear reviewer,
We thank you for your encouraging comments on our manuscript. After carefully considering all of the comments, we have performed the point-by-point revisions. Please find our responses to these comments/suggestions listed below.
The barley genome reference Morex v3 (Mascher et al. 2021) is well accepted in barley research community. I would suggest to modify the gene IDs MSTRG.29184 and MSTRG.29185 in main text and legends.
[Response 1] Thank you for this very important comment. In fact, in this study, as you said, the paired-end clean reads were mapped to the barley reference genome of barley cv. Morex (MorexV3) using HISAT2. 2.4 with “-rna-strandness RF” and other parameters set as the default. Generally speaking, genes found in the sequencing results but not included in the reference genome were defined as novel genes and represented by MSTRG. In order to better distinguish these novel genes in barley (Hordeum vulgare L.) from other species, we represented it by HvMSTRG in the revised paper. Accordingly, gene IDs MSTRG.29184 and MSTRG.29185 were modified to HvMSTRG.29184 and HvMSTRG.29185 in text and legends, respectively. Changes in the manuscript are marked in red.
Figure 6, the pictures (A-E) are pool for making interpretation. These panels are dispensable. Better to remove them.
[Response 2] Thank you for your advice. comment. In revised paper, we have removed pictures A-E in the figure 6 (line 255-263).
The reference genome for reads mapping is not given in Methods.
[Response 3] We apologize for the mistake. In revised paper, we have modified the related content in the manuscript (line 429-437).
The paired-end clean reads were mapped to the barley reference genome of barley cv. Morex (MorexV3) [40] using HISAT2. 2.4 with “-rna-strandness RF” and other parameters set as the default [41]. The mapped reads of each sample were assembled using StringTie v1.3.1 with a reference-based approach [42]. An FPKM (fragment per kilobase of transcript per million mapped reads) value was calculated for each transcription region to quantify its expression abundance and variations using StringTie software [43]. Genes found in this sequencing results but not included in the reference genome were defined as novel genes and represented by HvMSTRG.

Reviewer 2 Report
Dear Author,
your manuscript entitled “Characterization of glossy spike mutants and identification of candidate genes regulating cuticular wax synthesis in barley (Hordeum vulgare L.)” is well written, and the research is organized consistently with the objectives. Results have been properly reported and discussed.
Minor revisions are required as annotated on the attached file.
Sincerely

Author Response
Dear reviewer,
We thank you for your encouraging comments on our manuscript. After carefully considering all of the comments, we have performed the point-by-point revisions. Please find our responses to these comments in revised paper (marked in red). For example,
line 353-355: Briefly, plants were cultivated on acid soil and uniformly sown, fertilized and watered in greenhouse-controlled conditions under a 16 h light/8 h dark cycle with relative humidity 40-45%.
Reviewer 3 Report
This manuscript reports an eceriferum mutant of barley, the content of which is elegant in global bioinformatic and metabolinic analysis. However, genetic mapping of glossy spike mutant (cer-GN1) has not be performed. This is a critical defect because I can not evaluate whether this mutant is previously not characterized or not. Two candidate genes that were detected by RNA-seq is unclear about whether they correspond or not to the map location of the candidate gene. Molecular mapping information of the glossy mutant is the prerequisite for submission.
Author Response
Dear reviewer,
We thank you for your encouraging comments on our manuscript. After carefully considering all of the comments, we have performed the point-by-point revisions. Please find our responses to these comments/suggestions listed below.
This manuscript reports an eceriferum mutant of barley, the content of which is elegant in global bioinformatic and metabolinic analysis. However, genetic mapping of glossy spike mutant (cer-GN1) has not be performed. This is a critical defect because I can not evaluate whether this mutant is previously not characterized or not. Two candidate genes that were detected by RNA-seq is unclear about whether they correspond or not to the map location of the candidate gene. Molecular mapping information of the glossy mutant is the prerequisite for submission.
[Response] Thank you for this very important comment. RNA sequencing is not only useful to precisely determine gene expression profiles but also particularly powerful to detect novel genes/isoforms. Due to assembly errors and inaccurate prediction, the gene annotation of the reference genome may be incomplete. Transcriptome data contains transcripts expression information in a specific period. By reassembling these data, we can find out the genes that are not annotated in the original reference genome to a certain extent. After reconstituting the transcripts with stringtie software, the genes found in the sequencing results but not included in the reference genome (or reference gene set) were found and defined as novel genes.
In this study, we identified the Cer-GN1 barley (Hordeum vulgare L.) mutant generated by ethyl methanesulfonate mutagenesis with a glossy spike phenotype controlled by a single recessive nuclear gene. Using metabolomics and transcriptomics, we investigated the impact of cuticular wax composition and waxy regulatory genes on the loss of glaucous wax in the spikes of Cer-GN1. Finally, we identified two novel genes downregulated in Cer-GN1 (HvMSTRG.29184 and HvMSTRG.29185) that encode long-chain fatty acid omega-monooxygenase CYP704B1, which regulates the conversion of C16 palmitic acid to 16-hydroxyhexadecanoic acid. Their downregulation in Cer-GN1 reduced the synthesis of cuticular wax components and ultimately caused the loss of glaucous wax in the spikes. To further validate the results of this study, the expression levels of HvMSTRG.29184 and HvMSTRG.29185 in CER-GN1 and WT spike glume samples were analyzed at different growth stages 1, 4, 8, 12, and 16 days after heading (DAH) by qRT-PCR. HvM-STRG.29184 and HvMSTRG.29185 were significantly downregulated in CER-GN1 compared to WT, especially at 12 and 16 DAH. These results were consistent with the RNA-seq data, indicating that our analysis was reliable. These data suggested that HvMSTRG.29184 and HvMSTRG.29185 were candidate genes that could play roles in regulating cuticular wax synthesis. Of course, It is necessary to verify whether HvMSTRG.29184 and HvMSTRG.29185 truly encode a CYP704B1 that regulates the conversion of C16 palmitic acid to 16-hydroxyhexadecanoic acid in barley.
As you mentioned, genetic mapping of glossy spike mutant (cer-GN1) has not be performed and this work is very important for this study. Molecular mapping of the cuticular wax‑related candidate genes of HvMSTRG.29184 and HvMSTRG.29185 is the focus of our work. At present, on the one hand, we plan to carry out the cloning, transgene and gene editing research of these two genes; on the other hand, we plan to carry out the wax‑related genes mapping research by combining the bulk segregant analysis (BSA) and specific locus amplified fragment sequencing (SLAF-seq) strategy. We hope to get good results as soon as possible.
Round 2
Reviewer 3 Report
Almost no revision to my review comments has been made. This manuscript is obscure and is not informative to potential readers. Additional experiments are essential even to submit to other journals.